# Cisplatin Induces Apoptosis in Mouse Neonatal Testes Organ Culture

**DOI:** 10.3390/ijms232113360

**Published:** 2022-11-01

**Authors:** Hyun-Jung Park, Ji-Soo Kim, Ran Lee, Hyuk Song

**Affiliations:** 1Department of Stem Cell and Regenerative Biology, Konkuk University, 1 Hwayang-dong, Gwangjin-gu, Seoul 05029, Korea; 2Department of Animal Biotechnology, College of Life and Environment Science, Sangji University, Wonju-si 26339, Korea

**Keywords:** cisplatin, testis, meiosis, germ cell, oxygen stress

## Abstract

Chemotherapy is used for childhood cancer but may lead to infertility in patients. Spermatogonia stem cells are present in the testes of prepubertal boys, although they do not produce sperm at this age. Herein, we evaluated the toxicity of cisplatin, a known medicine for cancer treatment, in neonatal mouse testes using in vitro organ culture. Mouse testicular fragments (MTFs) derived from 5.5-d postpartum mouse testes were exposed to 1–10 μg/mL cisplatin. The results showed that cisplatin significantly downregulated the expression of germ cell marker genes, including differentiated and undifferentiated, in a dose-dependent manner. In particular, a high dose of cisplatin (10 μg/mL) led to germ cell depletion. In addition, the expression levels of the Sertoli cell marker gene, the number of SOX9+ Sertoli cells, and the levels of SOX9 protein were markedly decreased in cisplatin-treated MTFs compared to controls. The mRNA expression of steroidogenic enzyme-related genes significantly increased in cisplatin-treated MTFs, except for estrogen receptor 1 (Esr1). Consistently, 3β-hydroxysteroid dehydrogenase protein was also observed in the interstitial regions of cisplatin-treated MTFs. Altogether, our findings showed a significant impairment in germ cell development, Sertoli cell survival, and steroidogenesis in the MTFs of cisplatin-treated mice.

## 1. Introduction

Chemotherapy has been used for many years to treat childhood cancer. Although it kills cancer cells in the body, it can also damage healthy cells as a side effect. In particular, the preservation of gonadal function after chemotherapy is crucial for the long-term health of cancer survivors. The risk of infertility among childhood cancer survivors is concerning, with some patients almost certainly losing gonadal function [1]. Major factors determine the degree of testicular function damage among prepubertal boys receiving cytotoxic chemotherapy. The cumulative dose for administering alkylating agents such as cisplatin, cyclophosphamide, and busulfan is pronounced [2,3]. Cyclophosphamide administered orally at a cumulative dose of 0.7–52 g has been reported to cause gonadal damage and dysfunction in prepubertal boys [4]. In patients, testicular damage can affect the somatic or germ cells of the testes, and spermatogenesis can be disrupted after chemotherapy [5]. 

Germ cells and spermatogonial stem cells (SSCs) at early stages are present in the testes from approximately 2–3 months after birth [6,7]. During puberty, SSCs are maintained via self-renewal and differentiate to generate meiotic cells to produce sperm [8]. Other than germ cells, the seminiferous tubules of testes comprise Sertoli cells, which serve several crucial functions in the testis, including supporting spermatogenesis and germ survival. Sertoli cell proliferation occurs during fetal and early postnatal stages in humans. Normal maturation and proliferation of Sertoli cells are important to support spermatogenesis and male fertility [9,10]. Leydig cells are located within the interstitial area of the testes and produce key hormones such as testosterone and insulin-like growth factor (INSL3). Testosterone is required for normal testicular descent, masculinization, and spermatogenesis. Therefore, the health of each cell type in neonatal and prepubertal testes is important for sperm production after cancer treatment. 

Cisplatin is a major platinum compound that effectively treats several childhood cancers, such as osteosarcoma, hepatoblastoma, neuroblastoma, and germ cell tumors [11]. Despite having chemotherapeutic effects, cisplatin has several side effects on various organs, such as decreased renal function and increased red blood cell count [12,13,14]. In rodent testes, cisplatin induces testicular damage characterized by germ cell apoptosis, steroidogenic disorder, and Leydig cell dysfunction. In particular, disruption of spermatogenesis via inhibition of nucleic acid synthesis in germ cells and testosterone production leads to male infertility [15,16]. In addition, cisplatin leads to excessive production of reactive oxygen species (ROS), triggering cell death and DNA damage in both cancer [17] and normal cells [18]. In females, cisplatin also induces changes in the menstrual cycle and loss of primordial follicles via apoptosis by increasing ROS production and damage to DNA and mitochondria [19,20]. In males, cisplatin induces chronic testicular damage via testicular ROS production and endoplasmic reticulum stress. In particular, cisplatin treatment in mouse Sertoli cells showed increased ROS production [21]. However, studies on the mechanism of reproductive toxicity induced by cisplatin in neonatal testes are lacking. 

Testes organ cultures have been used for assessing toxicity to the reproductive system in both male and female rodents [22,23]. It can also be used to evaluate testis morphology and serve as a potential in vitro model for reproductive toxicity testing. Previously, we successfully conducted toxicity studies on the reproductive system using the neonatal testis organ culture method [24,25]. Moreover, other studies have also evaluated the toxicity of endocrine disruptor chemicals or toxins on the testes in an in vitro organ culture [26,27]. The organ culture method can reduce the number of test animals and effectively test chemicals and pharmaceuticals on reproductive organ development. 

In this study, we investigated the short-term effects of cisplatin on the development of neonatal testes and clarified the mechanism of the influence of cisplatin on neonatal testes using in vitro organ culture. To the best of our knowledge, this study is the first to examine the mechanism of cisplatin in neonatal testicular damage using a testis organ culture method. 

## 2. Results

### 2.1. Cisplatin Induced Testicular Damage in Organ Culture 

To assess the effect of cisplatin on neonatal testis development, mouse testicular fragments (MTFs) derived from 5.5-d-old neonatal testes were cultured for 4 d. Figure 1A shows the diagram of the experimental design. To determine whether cisplatin induced testicular damage, MTFs were cultured with cisplatin (1, 5, and 10 μg/mL) for 1 d, and the culture medium was replaced with a medium without cisplatin. Normally, germ cells in the testes undergo meiosis on postnatal day 8. SYCP3, a meiotic marker, was positive in MTF culture on day 2 (Figure 1B). Morphological changes in MTFs were observed from days 2 to 4 after in vitro culture (Figure 1C). Histological analysis with hematoxylin and eosin (H&E) staining showed partial testicular cell depletion in all MTFs exposed to 1, 5, and 10 μg/mL cisplatin. In addition, these morphological changes were more pronounced over time with higher cisplatin concentrations (Figure 1C). 

### 2.2. Effects of Cisplatin on Gene Expression of Germ Cell Markers in MTFs

Histological results showed that cisplatin induced testicular cell depletion in MTFs cultured in vitro. In addition, meiosis was initiated in germ cells of MTFs after 2 d of culturing. Therefore, we investigated whether cisplatin inhibited germ cell differentiation in MTF culture. The expression of germ cell marker genes, such as *Sycp3*, *Vasa*, *Sohlh2*, *Dazl*, *Sohlh1*, *Stra8*, *Gfrα-1*, and *Sall4*, decreased markedly in a dose-dependent manner compared to that in control after exposure to cisplatin for 4 d (Figure 1D). Both germ cell undifferentiated (*Gfrα-1* and *Sall4*) and differentiated (*Sycp3*, *Vasa*, *Sohlh2*, *Dazl*, *Sohlh1*, and *Stra-8*) marker genes were decreased by cisplatin. In other words, cisplatin toxicity to germ cells is not limited to specific germ cells during the developmental stage. The germ-cell-specific marker Vasa, undifferentiated germ cell marker Sall4, and differentiated germ cell marker SYCP3 were detected in cisplatin-exposed MTFs by immunohistochemistry. Results showed that Vasa+ cells were located on the basement membrane, and SYCP3+ cells were located in the middle region of the seminiferous tubules in all samples, including cisplatin-treated and -untreated MTFs. However, few Vasa+ and SYCP3+ cells were only present in MTFs exposed to high doses of cisplatin (CIS 10 μg/mL) (Figure 2A,B). An IgG isotype control was used as a negative control for immunostaining (Figure 2C). The comparison between the number of Vasa+, SYCP3+, and SALL4+ cells in control and cisplatin-treated MTFs revealed that the number of Vasa+, SYCP3+, and SALL4+ cells in cisplatin-treated MTFs were substantially decreased in a dose-dependent manner (Figure 2D). These results are consistent with the levels of SYCP3 and Vasa gene expression from the quantitative polymerase chain reaction (qPCR) results. In addition, we measured the protein levels of SYCP3 by immunostaining. Quantification by densitometry showed that the protein expression of SYCP3 was substantially reduced in MTFs exposed to 5 and 10 μg/mL cisplatin, whereas it did not differ between the control and 1 μg/mL cisplatin-treated MTFs (Figure 2E).

### 2.3. Effects of Cisplatin on Sertoli Cells in the MTF Culture

Seminiferous tubules contain an epithelium consisting of Sertoli cells, which is important for supporting germ cell development. Therefore, we investigated whether exposure to cisplatin affects Sertoli cell survival. The expression of the Sertoli cell marker genes *SOX9*, *AMH*, and *WT1* in MTFs was determined by qPCR. The levels of all Sertoli cell marker genes in cisplatin-exposed MTFs were substantially decreased in a dose-dependent manner (Figure 3A). In addition, SOX9 and vimentin proteins were detected in these samples. SOX9, typically expressed in Sertoli cells, was observed adjacent to the basement membrane of the tubules in control and 1 μg/mL cisplatin-treated MTFs. However, SOX9+ cells were rarely observed in 5 and 10 μg/mL cisplatin-treated MTFs (Figure 3B). The number of SOX9+ cells was significantly decreased in cisplatin-treated MTFs compared to that in control (Figure 3C). These results were consistent with the qPCR results. Additionally, the SOX9 protein levels were significantly decreased in cisplatin-treated MTFs, although the difference was not significant between the control and 1 μg/mL cisplatin-treated samples (Figure 3D).

### 2.4. Effects of Cisplatin on the Expression of Steroidogenic Genes in MTFs

Steroid hormones regulate diverse physiological functions during testicular development and are required for spermatogenesis. In particular, androgen is synthesized mainly by Leydig cells in the testes [28]. Steroidogenic enzymes are involved in steroid hormone biosynthesis [29]. Therefore, we investigated the expression of steroidogenic enzymes in cisplatin-treated MTFs to elucidate whether cisplatin affects steroidogenesis in Leydig cells of MTFs. The mRNA expression of the steroidogenic acute regulatory gene (*Star*), *Cyp11α1,* and *Cyp19α1* increased with increasing concentrations of cisplatin (Figure 4A). In addition, the expression of hormone receptor genes, such as androgen receptor (*Ar*), estrogen receptor 1 (Esr1), and luteinizing hormone receptor (*Lhr*), were detected by qPCR. The expression levels of *Ar* significantly increased in 5–10 μg/mL cisplatin-treated samples compared to the control. However, the gene expression of *Lhr* decreased in 10 μg/mL cisplatin-treated samples, and *Esr1* levels did not alter by cisplatin treatment (Figure 4B). Additionally, histological observation of MTFs showed that the Leydig cell population was present in the interstitial area of all samples (Figure 4C). Immunostaining of MTFs was performed using the 3β-hydroxysteroid dehydrogenase (3β-HSD) antibody, a Leydig cell-specific marker [30]. *3β-HSD*+ Leydig cells were detected in both cisplatin-treated and -untreated MTFs, similar to adult testis samples used as positive controls (Figure 4D). 

### 2.5. Cisplatin Induced Apoptosis-Related Protein Expression in MTFs

In our study, we revealed that cisplatin induced neonatal testicular damage in organ cultures. Therefore, we investigated the expression patterns of apoptosis-related genes and proteins in cisplatin-untreated and -treated MTFs. The expression levels of *BAX*, *BAD*, *FAS*, and *TRAF3* mRNAs, which play essential roles in the regulation of apoptosis, were analyzed. The expression of all genes was substantially higher at the high dose (10 μg/mL) of cisplatin exposure. The expression of *BAX*, *FAS*, and *TRAF3* was evident at low cisplatin doses (Figure 5A). Activation of caspase, a hallmark enzyme of apoptosis, was estimated by immunoblot analysis. The levels of pro-apoptotic proteins, such as Cleaved caspase-3, Cleaved-caspase-8, and cleaved PARP, in cisplatin-treated MTFs were substantially increased in a dose-dependent manner compared to the control, whereas Bcl-2 protein levels were lower in the cisplatin-treated group (Figure 5B). 

## 3. Discussion

All patients with cancer receive treatment that could affect their fertility. In particular, prepubertal and pubertal patients with cancer often experience infertility as a long-term complication of treatment. Prepuberty and puberty are crucial developmental periods and are sensitive to chemotherapies [31,32]. Sperm cryopreservation before chemotherapy can preserve fertility in male patients; however, fertility preservation in children has not been sufficiently elucidated [33,34,35].

In the present study, we investigated the toxic effects of cisplatin on MTFs derived from neonatal mice and cultured in vitro. In addition, the molecular mechanisms underlying the effects of cisplatin on MTFs were studied. Our results showed that cisplatin treatment of MTFs led to a significant reduction in germ cell number and the expression of germ cell marker genes. Similarly, the number of Sertoli cells and the expression of Sertoli cell maker genes were significantly decreased by cisplatin treatment. Leydig cells in cisplatin-treated MTFs were similar to that in controls; however, the expression of steroidogenic genes was increased by cisplatin treatment, indicating that cisplatin treatment led to hormone imbalance in cultured MTFs. Although there have been studies on the reproductive toxicity of cisplatin in male rodents, each study used a different treatment period, age of the model organism, daily dose, number of exposure cycles, and recovery period [36,37,38,39]. An in vivo toxicity test of cisplatin in mouse models showed that cisplatin exposure led to apoptosis of germ cells, including spermatogonia and spermatocytes, and decrease the number of spermatozoa and spermatids in seminiferous tubules of adult mouse testes [40]. 

Additionally, rodents exposed to cisplatin showed a decrease in the weight of both the testes and epididymides, as well as a reduction in testis size [41,42]. In some chronic cisplatin chemotherapy models, damage to the germ and spermatogenesis led to a lower number of spermatozoa, which was decreased by more than 90% compared to that in the untreated groups [43]. Furthermore, in vitro studies have also shown that C18-4 spermatogonia cells are sensitive to cisplatin, and cisplatin significantly increases DNA damage and telomere dysfunction in spermatogonia [44]. Another in vitro study showed that the activity of SSCs derived from 7–8 d postpartum (dpp) mouse testes was decreased by cisplatin treatment [45].

The effect of cisplatin could be similar or different between adult and neonatal testes, although adult testes are more complicated and involve a process of complete spermatogenesis. Similar to adult testes exposed to cisplatin in in vivo experiment, our result showed cisplatin-induced damage to various germ cells, including undifferentiated and differentiated germ cells. The testicular developmental period from 5.5 to 9.5 dpp in mice is important to define germ cell differentiation because meiotic cell division of germ cells in neonatal testes begins at approximately 8–10 dpp [46,47]. In our previous study, immunostaining with SYCP3 antibodies distinctly showed meiosis initiation at day 2 of culturing MTFs derived from 5.5 dpp mouse testes [24]. In humans, the seminiferous tubules of child testes consist of immature Sertoli cells and various spermatogonia, including A dark (Ad) and A paired (Apr) spermatogonia. Additionally, some tubules of the testes showed primary spermatocytes from 4 years of age [48]. This study focused on the effect of cisplatin on the progression of meiosis and undifferentiated germ cells. Ultimately, cisplatin-induced damage to germ cells was not specific to germ cell type, and meiotic cells were still observed in MTFs exposed to a high dose (10 μg/mL) of cisplatin. In contrast, our previous study showed that the toxicity of resmethrin was stronger in differentiated germ cells than in undifferentiated germ cells in MTF culture [24]. 

Cisplatin exposure on somatic cells such as Sertoli and Leydig cells of the testes has been reported to decrease inhibin-B and transferrin secretion of Sertoli cells [49,50]. Changes in Sertoli cell function may be responsible for cisplatin-induced damage to germ cells and spermatogenesis. Monsees et al. reported that the viability of Sertoli cells, derived from 18- to 21-d-old male rats, decreased by 15% after treatment with 100 μM cisplatin. Consistently, our result showed a decrease in Sertoli cell numbers in cisplatin-treated MTFs. Furthermore, the levels of *SOX9*, *WT1*, and *AMH* transcripts were lower in cisplatin-treated MTFs than in control. Wt1, a nuclear transcription factor, is important for fetal testes development, and knockdown of WT1 in postnatal Sertoli cells decreases sperm count because *WT1* promotes Sertoli cell–germ cell signaling networks that drive spermatogenesis [51,52]. Our study also showed changes in *WT1* gene expression in 10 μg/mL cisplatin-treated MTFs and massive germ cell death. The downregulation of the *WT1* gene expression in MTFs by cisplatin treatment may have indirectly affected germ cell damage. Sertoli cells are necessary for proper germ cell development and survival and orchestrate the process of spermatogenesis by supporting germ cell development in seminiferous tubules. Therefore, regardless of the mechanism of toxicity in testicular cells, chemicals such as toxicants, drugs, environmental pollen, and endocrine disruptor-induced Sertoli cell damage are associated with germ cell apoptosis or death in testes [53]. In an in vivo study, the administration of a single dose (5 mg/kg) of cisplatin to rats decreased the levels of testosterone in the blood [54]. In contrast, another study reported that the levels of gonadotropin and steroid hormones in serum rapidly increased upon cisplatin administration in rats [55,56]. The increase in steroidogenesis-related gene expression in cisplatin-treated MTFs cannot be elucidated because in vitro organ culture models cannot perfectly represent the endocrine system in the body. However, the change in steroidogenesis-related gene expression was undoubtedly induced by cisplatin treatment. An imbalance in steroidogenesis was observed in cisplatin-treated MTFs. A detailed study on the molecular mechanisms underlying the relationship between cisplatin and steroidogenesis in neonatal testes is required. 

Cisplatin induces cell death by forming inter- and intra-strand cross-linked DNA adducts and eventually inducing apoptosis [57]. In a mesothelioma cell line, cisplatin induces apoptotic cell death, along with mitochondrial depolarization, phosphatidylserine translocation, and caspase activation [58]. In acute promyelocytic leukemia (APL) cells, cisplatin exposure increased the expression of p53, AP-1, and p21, leading to cell cycle arrest and eventually causing apoptosis [59]. Similarly, our study showed that the expression levels of pro-apoptotic genes and proteins were remarkably increased in cisplatin-treated MTFs. Additionally, Appendix A shows the gene expression profile of ROS in MTFs after cisplatin treatment. The expression levels of genes known to be upregulated by ROS were distinctly increased in cisplatin-treated MTFs (Appendix A). These results support that cisplatin induces damage to neonatal testes through ROS-mediated apoptosis. The other study described that the molecular mechanism of platinum compound included cisplatin, carboplatin, and oxaliplatin. These platinum compounds induce damage of tumors via apoptotic signaling which mediated the death receptor mechanism as well as the mitochondrial pathway. In addition, several studies suggested that high levels of bax and p53 gene expression, inhibition of the JNK pathway, and higher concentration of glutathione and metallothioneins in tumor led to cell death by cisplatin treatment [60]. Based on our results, administration of cisplatin alone results in testicular cell damage. According to a previous study, combination therapies, which is the combination of platinum drug treatment with other drugs, radiation, and emerging gene therapy regimens, are used and have proven more effective to defect cancers for minimizing the cisplatin side effects of cancer therapy [60,61].

In summary, we investigated the toxicity of cisplatin in developing mouse testes using an organ culture method for the first time. However, a detailed study on the toxic mechanism of testicular damage by cisplatin is lacking and warrants further investigation. 

## 4. Materials and Methods

### 4.1. Animals

Two-day-old male mice and their mothers were obtained from Orient Bio (Seoul, Korea) and maintained for a few days. The mouse pups were used for organ culture 5.5 dpp. Mice were housed under a 12 h light:12 h dark photoperiod, and the temperature was maintained at 21 ± 1 °C. All animal experiments in this study were performed per the animal care policies of Konkuk University. Ethical approval for the project was granted by the Institutional Animal Care and Use Committee (IACUC) of Konkuk University (Register No. KU20139).

### 4.2. Organ Culture Method and Drug Treatment 

MTFs derived from 5 dpp neonates were used for in vitro culture, as described in previous studies [24,25]. Neonatal testes were removed from 10 male pups (5.5 dpp) for each experimental group. Initially, the testes were carefully dissected into 5–8 pieces of 2–3 mm in diameter using forceps. MTF sections (5–6 pieces) were randomly distributed onto 1.2% (*w/v*) agarose gel (Sigma-Aldrich, St. Louis, MO, USA) and placed in each well of a six-well plate (SPL Life Science, Gyeonggi-do, Korea). The samples were cultured in alpha-minimum essential medium (WelGENE, Daegu, Korea) supplemented with 10% (*v/v*) knockout serum replacement (Thermo Fisher Scientific, Waltham, MA, USA). Cisplatin was dissolved in phosphate-buffered saline (PBS) and diluted in the culture medium to final concentrations of 1, 5, and 10 μg/mL. MTFs were cultured with cisplatin in an incubator with 5% carbon dioxide in air at 34 °C. After 24 h of culture, samples were transferred to a medium without cisplatin, which was replaced once every 2 d. The cisplatin concentration was chosen based on previous studies [62,63], and the effects of the drug were determined across the range of cisplatin concentrations reported in patient serum concentrations of 4–10 μg/mL [64,65]. 

### 4.3. Histological and Immunohistochemical Analyses 

The MTFs were rinsed in PBS and fixed with 4% paraformaldehyde at room temperature (RT). The MTF samples were gradually dehydrated by increasing the alcohol concentration for 60 min. The dehydrated MTF tissues were cleared in xylene for 60 min and displaced by melting in paraffin for 2 h at 65 °C. Paraffin blocks were sectioned at 3 μm thickness using a microtome (Leica, Nussloch, Germany), and the sections were mounted on glass slides. For histological analysis, sample slides were stained with H&E for each of the five different areas of the tissue sections. For immunohistochemistry, the MTF sections were deparaffinized and rehydrated using xylene and 95–100% ethanol. Antigen was retrieved in 10 mM sodium citrate buffer, and the samples were heated for 15 min till the solution was boiled. Then, the tissues were soaked in PBS and blocked with PBS containing 0.01% Triton X-100 and 1% bovine serum albumin (BSA) for 30 min at 25 °C. Tissues were incubated overnight at 4 °C with diluted primary antibodies, such as SYCP3, VASA, SALL4, SOX9, vimentin, and 3β-HSD; the antibodies used for immunostaining are listed in Table 1. Following several washes with PBS, the tissues were incubated with secondary antibodies: Alexa Flour 488 goat anti-mouse IgG (Cat: A11001), Alexa Flour 594 donkey anti-rabbit IgG (Cat: R37119), and Alexa Flour 568 rabbit anti-goat (Cat: A11079), diluted 1:200 in 1% BSA in PBS for 1 h at RT. After several washes, the tissues were incubated with 1 μg/mL 6-diamidino-2-phenylindole (DAPI; Thermo Fisher Scientific) in PBS for 5 min, and coverslips were covered with mounting solution (DAKO, Carpinteria, CA, USA; S3025). Samples were analyzed using a Nikon E-800 fluorescence microscope (Nikon, Tokyo, Japan) with Motic Image Advanced 3.2 software. For quantitative analysis, more than five different MTF areas of each sample were used, and testicular cross-sections with more than 40 tubules from 5–6 sections were scored. 

### 4.4. RNA Isolation and qPCR

Total RNA from cultured MTFs was extracted using a Qiagen RNeasy Mini Kit (Cat: 74106; Qiagen, Hilden, Germany) with on-column DNase treatment (Cat: 79254; Qiagen), according to the manufacturer’s instructions. Approximately 3–5 testes fragments contained 1 µg RNA after 4 d of culture. cDNA was synthesized from 1 µg of total RNA using SuperScript Ⅲ Reverse Transcriptase (Invitrogen, Carlsbad, CA, USA) with Oligo(dT30)-primer and 1 µg RNA, following the manufacturer’s instructions. Primers used in this study are listed in Table 2. Denaturation and polymerase activation were performed at 94 °C for 1 min, followed by 40 cycles of 94 °C for 10 s, 57 °C for 10 s, and 72 °C for 20 s. Data were analyzed using the comparative Ct method [66], and *Gapdh* was used as the control gene. After normalization to *Gapdh* levels, which were reflected in the ΔCt values, the relative quantification (RQ) of the fold-change for each treatment compared with that of the reference control was determined using the following equation:RQ = 2^(−ΔCt)^/2^(−ΔCt reference)^.(1)

The mean RQ and SEM values were plotted on a log_2_ scale. In addition, ROS-related gene expression profiling was performed using the AccuTarget^TM^ mouse oxidative stress qPCR screening kit (Bioneer, Daejeon, Korea).

### 4.5. Western Blot Analysis 

Proteins samples from cultured MTFs were extracted using ice-cold RIPA buffer (Thermo Fisher Scientific) containing protease inhibitors (Roche, Indianapolis, IN, USA). Total protein was quantified using a BCA Protein Assay Kit (Pierce Biotechnology, Rockford, IL, USA; #23 277). Approximately 4–6 testes fragments contained 30 µg of protein. Samples were loaded with an equal volume of protein into wells containing 10% or 15% SDS-PAGE gels, and the proteins were transferred from the gel to PVDF membranes. Membranes were incubated overnight at 4 °C with primary antibodies diluted in Tris-buffered saline with Tween-20 (TBST, 20 mM Tris-HCl, pH 7.5, 150 mM NaCl, and 0.1% Tween-20) containing 1% BSA. The primary antibodies used in the study are listed in Table 1. After three washes in TBST (5 min each), the membrane was incubated for 1 h with 1:20000 dilution of anti-mouse and anti-rabbit IgG and horseradish peroxidase (HRP)-linked antibody (Thermo Fisher Scientific) in TBST containing 1% BSA. After washing for 1 h, the blots were visualized using Pierce ECL Western blotting substrate (Thermo Fisher Scientific; No. 34580) and HyBlot CL autoradiography film (Denville Scientific, Metuchen, NJ, USA; No. E3018). β-actin was used as the control for normalization.

### 4.6. Statistical Analysis

SPSS statistical package ver. 15.0 for Windows (IBM Corp., Somers, NY, USA) was used for data analysis. All data are expressed as mean ± standard error value from at least three independent experiments and were evaluated using one-way analysis of variance (ANOVA), followed by Tukey’s honest significance test. Values of * *p* < 0.05 and ** *p* < 0.01 were considered statistically significant.

## Figures and Tables

**Figure 1 ijms-23-13360-f001:**
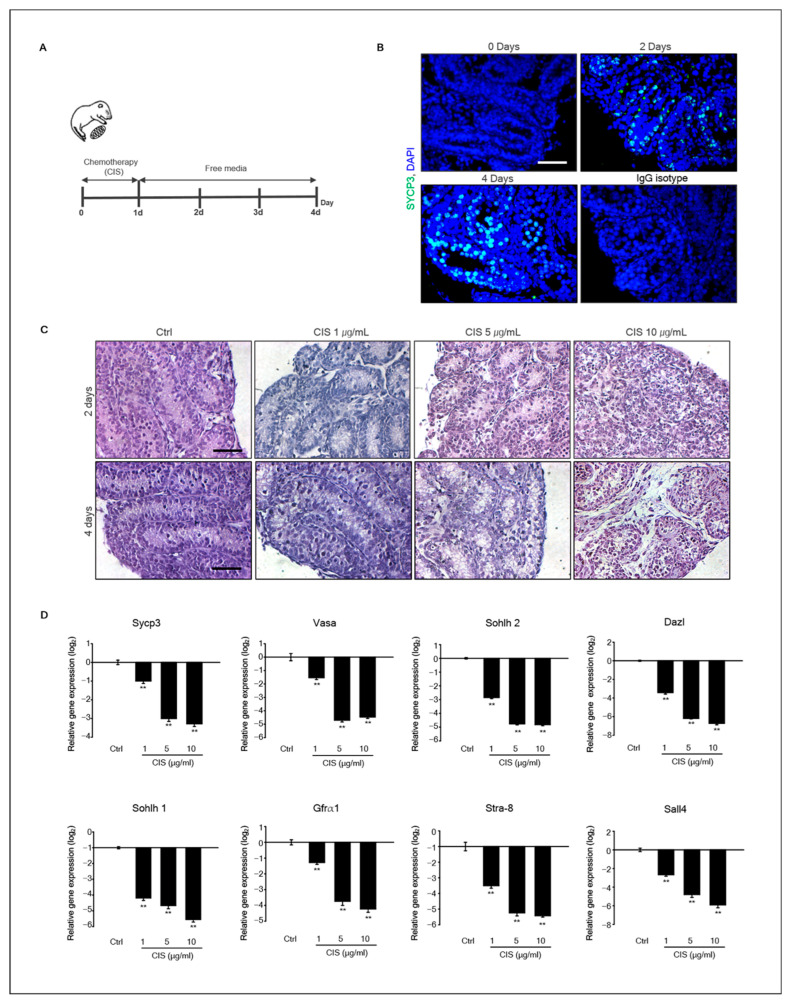
**Morphological changes in MTFs induced by cisplatin and gene expression of germ cell marker in MTFs.** (**A**) Experimental design diagram. MTFs derived from 5.5 d postpartum mice were cultured with cisplatin (1, 5, and 10 μg/mL) for 1 d, and the medium was replaced with a cisplatin-free medium for 3 d. (**B**) Immunostaining of *SYCP3* on cultured MTFs during 2–4 d. (**C**) Histological assessment using hematoxylin and eosin staining of 1–10 μg/mL cisplatin-exposed MTFs cultured for 2, 3, and 4 d. Scale bars = 50 μm; each image was observed at the same magnification. (**D**) Gene expression of germ cell markers in cisplatin-exposed MTFs. The expression levels of germ cell marker genes, *Sycp3*, *Vasa*, *Sohlh2*, *Dazl*, *Sohlh1*, and *Stra8*, and undifferentiated germ cell markers *Gfra1* and *Sall4* in the MTFs were determined using quantitative polymerase chain reaction (qPCR). Relative quantification of mRNA is shown using the mean and the standard error of the mean (*n* = 5) at the log_2_ scale. Data were analyzed using Student’s unpaired *t*-test, ** *p* < 0.01.

**Figure 2 ijms-23-13360-f002:**
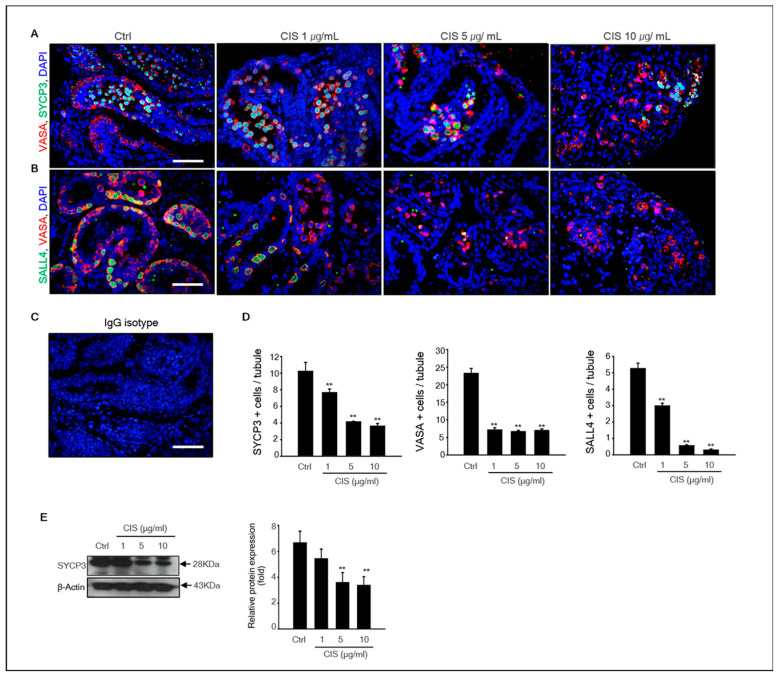
**Cisplatin toxicity on germ cell development**. (**A**) Meiotic and total germ cells were co-treated with SYCP3 and Vasa antibodies to confirm the occurrence of meiosis of germ cells in 1, 5, and 10 μg/mL cisplatin-exposed MTFs. (**B**) Undifferentiated and total germ cells co-treated with SALL4 and Vasa antibodies in cisplatin-exposed MTFs. (**C**) IgG isotype as a negative control. The average number of each. Scale bars = 50 μm; each image was observed at the same magnification. (**D**) SYCP3, Vasa, and SALL4-positive cells in the tubules were calculated by immunostaining, and at least 40 tubules were scored for each MTF. (**E**) Immunoblot analysis for SYCP3 expression in cisplatin-exposed MTFs. *β*-actin was used as a loading control. Quantitative analysis of each protein levels. Data shown using the mean and the standard error of the mean (*n* = 5), ** *p* < 0.01.

**Figure 3 ijms-23-13360-f003:**
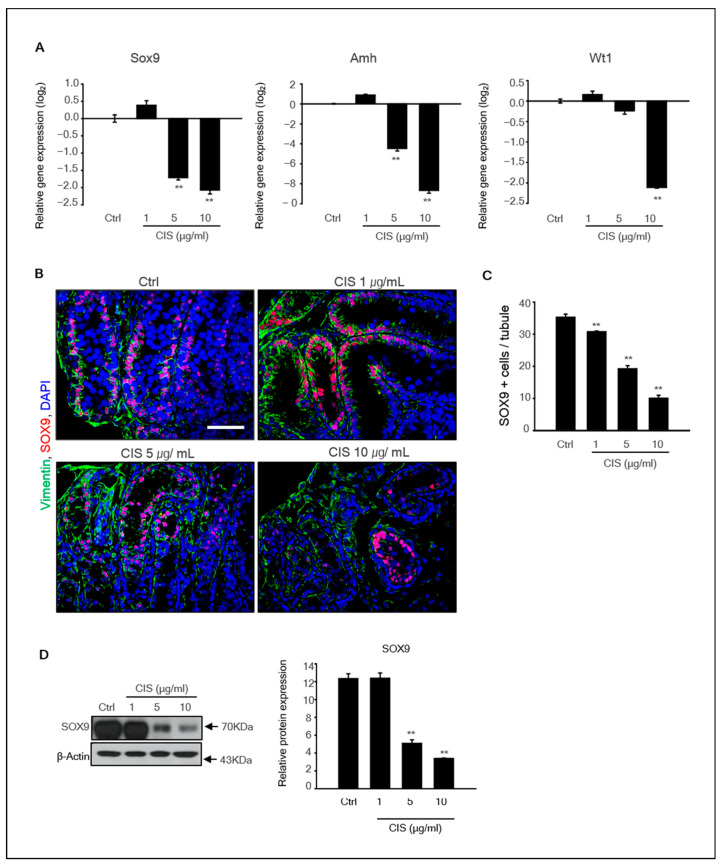
**Effects of cisplatin on Sertoli cells in in vitro MTF culture.** (**A**) The expression levels of Sertoli cell marker genes *SOX9*, *AMH*, and *WT1* in cisplatin-treated MTFs were determined by qPCR. Relative quantification of mRNA is shown as the mean and standard error of the mean (*n* = 5) at the log_2_ scale. (**B**) Double immunostaining of the MTFs section was performed using SOX9 and vimentin antibodies, which are known as Sertoli cell markers. Scale bars = 50 µm. All images were acquired at the same magnification. (**C**) The number of SOX9+ cells in tubules was counted based on the immunostaining image. At least 40 tubules were scored for each MTF. The data are presented as mean and standard error of the mean (*n* = 5), ** *p* < 0.01. (**D**) The expression of SOX9 protein was determined by immunoblot analysis in cisplatin-treated and -untreated MTFs. Quantitative analysis of the SOX9 protein is shown as the mean and standard error of the mean (*n* = 5) at the log_2_ scale. ** *p* < 0.01.

**Figure 4 ijms-23-13360-f004:**
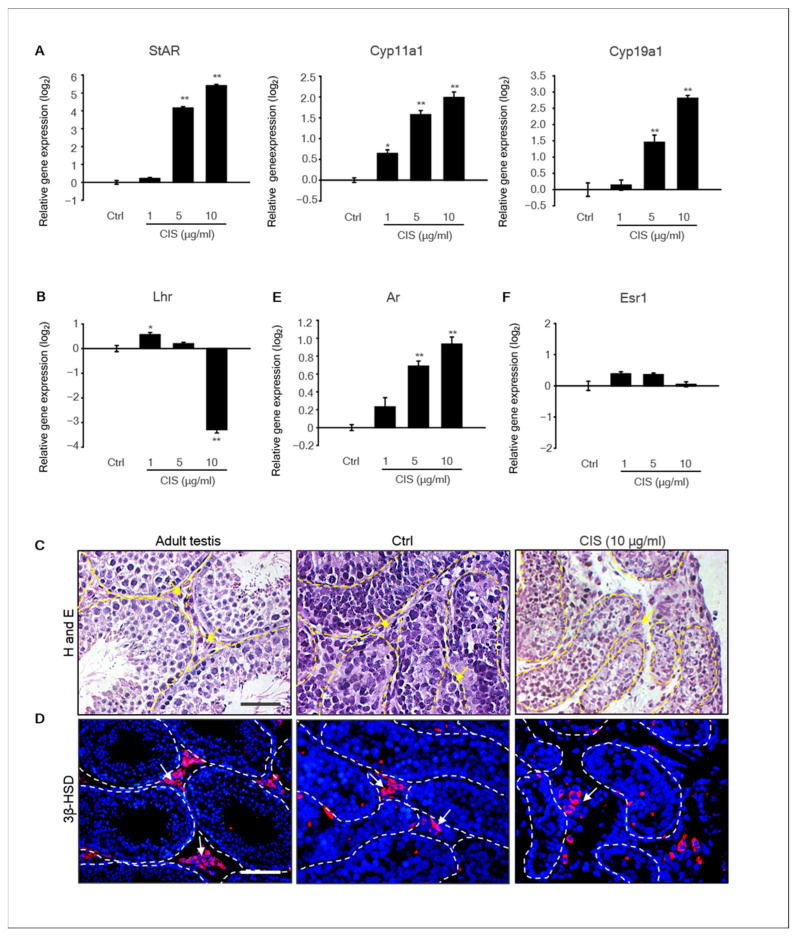
**Expression of the steroidogenic marker in the MTF after cisplatin treatment.** (**A**) The expression levels of the steroidogenic marker genes *Star*, *Cyp11α1*, and *Cyp19a1* and the expression levels of hormone receptors (**B**) *Lhr,* (**E**) *Ar*, and (**F**) *Esr1* in cisplatin-treated MTFs were determined using qPCR. The relative quantification of mRNA expression is shown as the mean and the standard error of the mean (*n* = 5) at the log_2_ scale. * *p* < 0.05, ** *p* < 0.01. (**C**) The histological data revealed that Leydig cells were present in the interstitial region of MTFs exposed to cisplatin, non-treated MTFs, and adult testes (yellow arrowhead). (**D**) Steroidogenic marker, 3β-HSD protein, was detected in cisplatin-treated (10 μg/mL) and -untreated MTFs by immunostaining (white arrowhead). Scale bars = 50 µm. All images were acquired at the same magnification.

**Figure 5 ijms-23-13360-f005:**
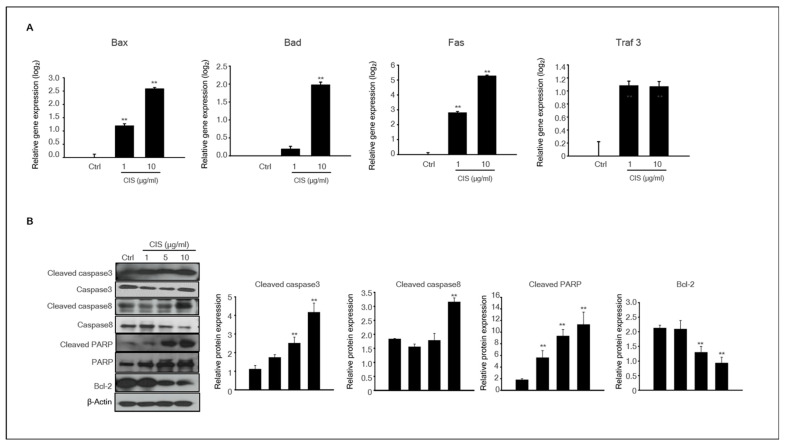
**Expression of apoptosis-related genes and proteins in cisplatin-treated MTFs.** (**A**) The expression levels of the apoptotic genes Bax, Bad, Fas, and Traf3 in the MTFs were determined by qPCR after treatments with 1 and 10 μg/mL cisplatin. Data are represented as the mean and the standard error of the mean (*n* = 5) at the log_2_ scale. ** *p* < 0.01 (**B**) Pro-apoptotic proteins such as cleaved caspase-3, cleaved caspase-8, cleaved PARP, and Bcl-2 were detected in 0–10 μg/mL cisplatin-treated MTFs by immunoblotting. *β*-actin was used as a loading control. Quantitative analysis of each protein level. Data are shown using the mean and the standard error of the mean (*n* = 5), ** *p* < 0.01.

**Table 1 ijms-23-13360-t001:** List of antibodies used for immunostaining.

Antibody	Company	Catalog Number	Dilution (Usage)
SYCP3	Abcam	ab97672	1:200 (IHC)
VASA	Abcam	ab13840	1:200 (IHC)
SALL4	Abcam	ab57577	1:200 (IHC)
Vimentin	Santa Cruz Biotech	sc-373717	1:100 (IHC)
SOX9	Abcam	ab185966	1:200 (IHC)
3βHSD	Santa Cruz Biotech	sc-30820	1:100 (IHC)
β-actin	Santa Cruz Biotech	sc-47778	1:2000 (WB)
Bcl-2	Cell signaling	#3498	1:2000 (WB)
Cleaved PARP	Cell signaling	#9541	1:2000 (WB)
Cleaved caspase-3	Cell signaling	#9664	1:2000 (WB)
Cleaved caspase-8	Cell signaling	#8592	1:2000 (WB)
Caspase-3	Cell signaling	#9662	1:2000 (WB)
Caspase-8	Cell signaling	#9746	1:2000 (WB)
PARP	Cell signaling	#9532	1:2000 (WB)

**Table 2 ijms-23-13360-t002:** Primers used for reverse-transcription polymerase chain reaction using mouse cDNA.

Gene	Forward Primer	Reverse Primer
Sycp3	5′-CAGATGCTTCGAGGGTGTG-3′	5′-AAGGTGGCTTCCCAGATTTC-3′
Vasa	5′-CCGCATGGCTAGAAGAGATT-3	5′-TTCCTCGTGTCAACAGATGC-3
Sohlh1	5′-CATCTGCTGTGTCTCGGGTA-3′	5′-GCTGGAAGACTCTGGCTCAC-3′
Sohlh2	5′-TGAGACGAGAACGCATCAAG-3′	5′-CCTCTGTGATGTGGCTGAGA-3′
Dazl	5′-GTCGAAGGGCTATGGATTTG-3′	5′-ACGTGGCTGCACATGATAAG-3
Gfra 1	5′-CCTGGATTTGCTGATGTCG -3′	5′-GCTGAAGTTGGTTTCCTTGC-3′
Stra-8	5′-CTGAGGCTGTTGGACCAGAT-3′	5′-GCAACAGAGTGGAGGAGGAG-3′
Sall4	5′-CCTCGGTGTTAGATGTCAAGG-3′	5′-GGGCACACGTAAGGTCTCTC-3′
Sox9	5′-AGTACCCGCATCTGCACAAC-3′	5′-TACTTGTAATCGGGGTGGTCT-3′
Wt1	5′-ATCCCAGGCAGGAAAGTGTG-3′	5′-GTGCTGTCTTGGAAGTCGGA-3′
Amh	5′-CCTGGAGGAAGTGACATGG-3′	5′-CAGGGTAGAGCACCAGCAG-3′
AR	5′-GGCGGTCCTTCACTAATGTC-3′	5′-GACAGGTGCCTCATCCTCAC -3′
Esr1	5′-GCACAAGCGTCAGAGAGATG-3′	5′-AGGACAAGGCAGGGCTATTC-3′
Cyp19α1	5′-TTGAGACGATTCCAGGTGAAG-3′	5′-ATTTCCACAAGGTGCCTGTC-3′
Cyp11α1	5′-GACAATGGTTGGCTAAACCTG-3′	5′-GGGTCCACGATGTAAACTGAC-3′
Lhr	5′-GGCCATCCTCATCTTCACAG-3′	5′-TTGGCACAAGAATTGACAGG-3′
Star	5′-TGGGCATACTCAACAACCAG-3′	5′-GTCTACCACCACCTCCAAGC-3ʹ
Bad	5′-GCCCTAGGCTTGAGGAAGTC-3′	5′-GGCTCAAACTCTGGGATCTG-3′
Bax	5′-GCTGACATGTTTGCTGATGG-3′	5′-GATCAGCTCGGGCACTTTAG-3′
Fas	5′-CTGATCCTCATTCCCGTACC-3′	5′-ATCATTGGCACCTCTTCAGC -3′
Traf3	5′-AGAGTGAGTTGAGTGCACACTT-3′	5′-TACCGCGGAGCTGGCCTCAT-3′

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
