# Peer review of "Cisplatin Induces Apoptosis in Mouse Neonatal Testes Organ Culture"

_ijms, 2022, doi:10.3390/ijms232113360_

Round 1
Reviewer 1 Report
Although the effects of cisplatin on mouse and rat testis had already been investigated in vivo, the manuscript by Park and colleagues provides the first evidence of its toxicity in neonatal tissue, using testes organ cultures as a model system. They further provide mechanistic effects for the toxicity of cisplatin in neonatal testicular damage. This study further supports the validity of using neonatal testes organ cultures for toxicity assays. The experimental design and methodology used has been clearly presented, and they are both appropriate. The manuscript is well written too, and easy to follow. The data presented are solid, and they follow a logical order. I would thus be supportive of this manuscript to be accepted in its current form.
Author Response
Comments and Suggestions for Authors
Reviewer1)
Although the effects of cisplatin on mouse and rat testis had already been investigated in vivo, the manuscript by Park and colleagues provides the first evidence of its toxicity in neonatal tissue, using testes organ cultures as a model system. They further provide mechanistic effects for the toxicity of cisplatin in neonatal testicular damage. This study further supports the validity of using neonatal testes organ cultures for toxicity assays. The experimental design and methodology used has been clearly presented, and they are both appropriate. The manuscript is well written too, and easy to follow. The data presented are solid, and they follow a logical order. I would thus be supportive of this manuscript to be accepted in its current form.
- Our response: Thanks you for taking the time for your attention and review
Reviewer 2 Report
It is clear for long time that cisplatin is highly toxic to cancer cells and others especially when used alone due to its intercalating DNA properties. It is often associated in cocktails as noted in literature.
In the text the authors do not named the different cells associated with spermiogenesis and spermatocytogenesis... i.e., Lines 240 and afterwards
Basic references are missing with other toxicities as noted in text attached.
Please amend your interesting research study.
Cisplatin induces apoptosis in mouse neonatal testes organ culture
L 240 spermatogonia, spermatocytes, and spermatids of adult mouse testes, and deceased
spermatogenesis [40].
L 308 However, a detailed study on the toxic mechanism of testicular damage by cisplatin is lacking
L 309 and warrants further investigation.
Dasari S, Tchounwou PB. Cisplatin in cancer therapy: molecular mechanisms of action. Eur J Pharmacol. 2014 Oct 5;740:364-78. doi: 10.1016/j.ejphar.2014.07.025.
Ghosh S. Cisplatin: The first metal based anticancer drug. Bioorg Chem. 2019 Jul; 88:102925. doi: 10.1016/j.bioorg.2019.102925.
‘ However, side effects and drug resistance are the two inherent challenges of cisplatin which limit its application and effectiveness. Reduction of drug accumulation inside cancer cells, inactivation of drug by reacting with glutathione and metallothioneins and faster repairing of DNA lesions are responsible for cisplatin resistance. To minimize cisplatin side effects and resistance, combination therapies are used and have proven more effective to defect cancers.’
Boulikas T, Vougiouka M. Cisplatin and platinum drugs at the molecular level. (Review). Oncol Rep. 2003 Nov-Dec;10(6):1663-82. PMID: 14534679.
This review summarizes the molecular mechanisms of platinum compounds for DNA damage, DNA repair and induction of apoptosis via activation or modulation of signaling pathways and explores the basis of platinum resistance. Cisplatin, carboplatin, oxaliplatin and most other platinum compounds induce damage to tumors via induction of apoptosis; this is mediated by activation of signal transduction leading to the death receptor mechanisms as well as mitochondrial pathways. Apoptosis is responsible for the characteristic nephrotoxicity, ototoxicity and most other toxicities of the drugs. The major limitation in the clinical applications of cisplatin has been the development of cisplatin resistance by tumors. Mechanisms explaining cisplatin resistance include the reduction in cisplatin accumulation inside cancer cells because of barriers across the cell membrane, the faster repair of cisplatin adducts, the modulation of apoptotic pathways in various cells, the upregulation in transcription factors, the loss of p53 and other protein functions and a higher concentration of glutathione and metallothioneins in some type of tumors. A number of experimental strategies to overcome cisplatin resistance are at the preclinical or clinical level such as introduction of the bax gene, inhibition of the JNK pathway, introduction of a functional p53 gene, treatment of tumors with aldose reductase inhibitors and others. Particularly important are combinations of platinum drug treatments with other drugs, radiation, and the emerging gene therapy regimens.
Author Response
Reviewer2)
Comments and Suggestions for Authors
It is clear for long time that cisplatin is highly toxic to cancer cells and others especially when used alone due to its intercalating DNA properties. It is often associated in cocktails as noted in literature.
In the text the authors do not named the different cells associated with spermiogenesis and spermatocytogenesis... i.e., Lines 240 and afterwards.
- Our response: we exactly corrected base on reference.
Basic references are missing with other toxicities as noted in text attached. Please amend your interesting research study.
- Our response: Thanks you for taking the time for your attention and review. We agreed your comments. So, we revised manuscripts. Especially the part of discussion follows your comments. We added and corrected based on your attached text (blue highlight).